# Solar Charging of Electric Vehicles: Experimental Results

**Bruno Robisson** [1,*]**, Sylvain Guillemin** [2]**, Laurie Marchadier** [1]**, Gérald Vignal** [3] **and Alexandre Mignonac** [1]

[1] Commissariat à l'Energie Atomique et aux Energies Alternatives (CEA), CEA Cadarache, 13108 Saint-Paul-lèz-Durance, France; laurie.marchadier@cea.fr (L.M.); alexandre.mignonac@cea.fr (A.M.)
[2] Univ. Grenoble Alpes, CEA, Liten, INES, ITE INES.2S, 50 Av. du Lac Léman, 73370 Le Bourget-du-Lac, France; sylvain.guillemin@cea.fr
[3] Réseau de Transport d'Electricité (RTE), 7C Place du Dôme, 92073 Paris, France; gerald.vignal@rte-france.com
\* Correspondence: bruno.robisson@cea.fr

**Abstract:** Sales of electric vehicles, for commercial use and personal use, keep rising. In parallel of the development of the associated Electric Vehicle Charging Infrastructure (EVCI), systems for controlling the charging of EVs will have to be developed in order to reduce the impact of such a development on the power grid. In this paper, we present a supervision system that controls the electric vehicle charging of employees of CEA Cadarache research center. The EVCI of Cadarache, set up in 2016, is constituted of more than 80 22-kW AC charging points spread over 30 zones. This EVCI currently supplies more than 376 vehicles including taxis, service vehicles as well as employees' vehicles. This infrastructure is one of the largest private EVCIs in the region. The supervision system controls electric vehicle (EV) charging in real-time according to two objectives: respecting user preferences, by fully charging the EV battery, and synchronizing the power consumption of a fraction of the EVCI, i.e., 24 charging points, with the power production of a solar photovoltaic plant. This paper details the supervision system that is used to carry out these experiments and presents experimental results. These results show that it is technically feasible to increase (up to 60 percentage points) the self-production ratio while satisfying EV users.

**Keywords:** electric vehicles (EVs); photovoltaic (PV) production; smart charging; planning algorithm; field test; self-production

## 1. Introduction

The European Commission plans to ban the sale of new non zero-emission commercial and passenger light vehicles from 2035 [1]. This decision will accelerate the development of full Battery Electric Vehicles (BEV), which represented 9% (880k units) of sales of light vehicles in Europe over 2021 [2]. This growth needs to be supported by a rapid development of the Electric Vehicle Charging Infrastructure (EVCI). By the end of 2021, there were 225,000 charging points available in the European Union [3] but by 2030, a total of four million charging points are expected [4], for around 34 million BEVs and 14 million Plug-in Hybrid Electric Vehicles (PHEVs). The impacts of this growth on the power grid must to be carefully anticipated: greater electricity production will be needed, and electricity transmission and distribution networks will have to be improved [5,6]. For example, the French transmission system operator, named RTE, has estimated in a recent study [7] that, in France, the electricity production will have to be increased by 100 TWh in 2050 in a "reference scenario", considering that 95% of light vehicles and 21% of duty vehicles are electrified. This energy will represent approximately 15% of the projected electricity consumption in this scenario. Nonetheless, in the same study, RTE concludes that the control of the charge of electric vehicles (EVs) is a "no regret" solution from a technical and economic standpoint, i.e., it is profitable in all situations.

Many control algorithms and associated systems have been proposed in the literature [8,9]. For example, authors propose to provide primary reserve [10,11], i.e., to change

the charging power of the EV according to the frequency deviation of the network, or to limit EV load to the power network capacity available [12–14]. Other authors propose to provide reactive power [15] or to minimize the effects on the grid of the rapid photovoltaics (PV) production output fluctuations due to cloud transients [16] or even to balance wind energy [17]. Other control algorithms aim to minimize the charging cost for the user [18], for a fleet manager [19] or for a parking operator [20]. Some studies also analyze the EV charging when integrated in a microgrid with the consumption of household or/and commercial buildings, solar carport and/or solar plant and eventually an energy storage system [21–27]. All these articles are either based on synthetic data (i.e., generated from mathematical models only) or use real data (i.e., obtained from physical measures) as inputs for simulations. Such real data are obtained from laboratories or from field scale demonstrators. Marinelli and al. propose a review of projects that deal with such demonstrators in Europe [28]. Descriptions of other demonstrators may also be found in [29–34].

Nonetheless, the literature lacks papers or public reports that describe in detail experiments that control the charge of EVs in a field scale demonstrator, except in the following four studies. In [30], the authors described tests that have been conducted from January 2017 to December 2018 in the UK with more than 600 EV drivers involved. During one of these tests, called "Trial 3", the users were financially incentivized to let the system charge their EVs outside the peak hours. The results show that the incentive, combined with smart charging, had a significant impact on drivers' behavior. In [35], the authors describe an experimental set-up with commercial EVs and two commercial unidirectional charging stations in the test-site of University Campus Lyngby. The authors demonstrate that it is technically feasible to control the unidirectional charging of EVs to provide primary frequency regulation. The project ChargeForward [31,36] has managed a set of more than 400 EV-driving households (approximately 250–300 at a given time in the project) that have participated in real-world experiments in the San Francisco Bay area. Several use cases of EV grid integration were studied. One of them, called "Earth Week Renewable Energy Use Case", consisted of inciting the participants to charge in the middle of the day to use the excess of solar energy. This use case lasted one week in 2018. The project showed that 55% of the charging powers came from renewable energy, compared to the national average of 23%. In [37], the authors describe the results of an experiment that was conducted during a year on a test-site constituted with six 22-kW AC charging stations. They show that their algorithm ensures a fair distribution of the charging power between the six charging stations even when the grid connection only allows two EVs to charge concurrently. In [32], the same kind of test was realized but on a larger scale, i.e., an EVCI constituted of more than 1000 public charging points in the Netherlands. The researchers showed that it was possible to limit the speed of the charging of the EV in order not to exceed the power network capacity available.

This paper describes the set-up and the results of smart charging tests on a CEA Cadarache site. During these tests, we controlled the charge of EVs in order to recharge the EV batteries and to maximize the self-production rate of the system composed of 24 22-kW AC charging stations and a 160-kWp photovoltaics (PV) plant. These tests involved more than 300 users that own more than 40 different EV models.

The remainder of the paper is organized as follows. The experimental set-up is detailed in Section 2, with a part that focuses more on the control algorithm. The main results obtained over a four-month period and the lessons learned are presented in Section 3. Section 4 presents the conclusion of these experiments and discusses the perspectives.

## 2. Context of the Experiments

### 2.1. R&D Center of Cadarache

The experiments take place in the Cadarache research center of the French Alternative Energies and Atomic Energy Commission, or CEA (the French acronym for "Commissariat à l'Énergie Atomique et aux Énergies Alternatives"). It is a 60 year-old research center located near Aix-en-Provence. This center is spread over 1600 Ha of which 900 Ha are

fenced (by a 22 km fence). Cadarache center consists of 480 buildings, including office buildings but also research laboratories. The center is directly connected to the electricity network. The CEA manages its own water network, heat network, and medium voltage electricity distribution network (made up of 18 15-kV loops). The research center plays the role of Distribution System Operator (DSO). A public lighting network and an EVCI are connected to this power network. Two thousand five hundred CEA employees work at the Cadarache site, as well as partner companies. In total, around 5000 people work at Cadarache. The CEA offers a private bus service to its employees for commuting. Thus, Cadarache center can be seen as a small town privately owned and managed by the CEA.

The CEA conducts research on solar thermal energy and on solar photovoltaics. There are therefore PV solar plants installed on the Cadarache site in two different locations, called the internal solar platform and the Mégasol platform. On the internal platform, the CEA tests and evaluates innovative PV solutions ranging in size from modules to a few tens of kilowatts systems. On Mégasol's platform, the CEA tests these innovative solutions on four PV plants totalizing 12 MWp, owned by industrial partners.

## 2.2. EVCI

The CEA's EVCI was set-up during the summer of 2016. It involves 40 Diva-type terminals, produced and installed by G$^2$Mobility, which was bought out by TotalEnergies in 2018. Each Diva terminal has two 22-kW AC charging points. Each of these charging points has a Type 2 socket for mode 3 connection and a Type E socket for mode 1 and 2 connections. These charging stations have been installed alone or in groups of up to four Diva terminals, to create 30 charging stations, spread out through the whole center. Each charging station has an IoT gateway embedded that enables communication through 3G networks by using Open Charge Point Protocol (OCPP) commands.

The CEA maintains and operates this EVCI. The CEA therefore plays the role of Charging Point Operator (CPO). The user must use a badge to authorize charging of the EV. To obtain his badge, the user has to be registered and has to specify his contact details, such as mobile phone number, as well as his car's model. Thus, the CEA assumes the role of e-Mobility Service Provider (e-MSP).

## 2.3. EVCI Charging History

Before performing our experiments, we have collected data from an OCPP supervisor for 4 years (from June 2016 to June 2020). During this period, 17,045 charging sessions were recorded, totalizing 253 MWh of energy consumption. The mean duration and the mean energy consumption of each charging session were respectively 12 h and 14.8 kWh. We also have computed the histograms of the start and end times of the charging sessions. Figure 1 reports the results. The X-axis represents the local time. The start times of the charging sessions are in blue and the end times are in orange. It appears that there are three main periods when the charging sessions start: mostly at the beginning of the working day (around eight in the morning), then at lunchtime when people charge their car at the business restaurant, and at the end of the afternoon when service cars come back from business trips. There are also three main periods for the end of charging sessions. The first one is at 9 am when the service cars that have been connected the day before are disconnected to be used for business trips, the second one is at lunchtime and the last period is at the end of the working day (around 5–6 pm) when employees leave the center.

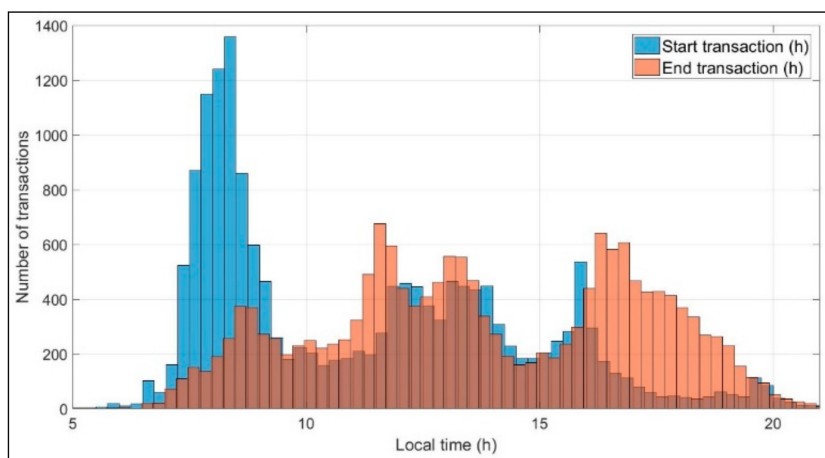

**Figure 1.** Histograms of start and end times of charging sessions.

### 2.4. EVCI Users

As at 1 February 2022, 376 RFID badges were issued for EV (including PHEV) owners. The number of badges is steadily increasing by around 100 badges per year. Two hundred and sixty cars are used by employees for their personal use. There are also 84 service cars. Cadarache site has also set up a taxi service provided by three Zoé cars. Besides, the CEA authorizes external companies to charge their 29 EVs.

There are three involvement levels of the users in our experimentations:

- Seventy-five EV owners (i.e., about 78% of the volunteer experimenters) agree to give us control over the charging power of their charging sessions. They have also consented to send information about the State of Charge (SOC) of their battery and their forecast departure time. They also provide details on their car's features (capacity of the battery and maximum power of the onboard charger). The set of these cars is further called "PControlled".
- Twenty-one employees have not given us control over the charge of their car, but have agreed to send us information about their SOC and their forecast departure time. They also give us details on their car. The set of these cars is further called "PUncontrolled".
- The other users do not participate to the experiments. They are named "Others".

The union of the first two categories of people is further called "Participants". The union of the last two categories is called "Uncontrolled".

In terms of car models, there is a very clear predominance of Renault Zoé. Such a model represents 38% of all the EVs. There are also, among others, 10% of Peugeot e208, 7% of Renault Twingo and 8% of Tesla (Model 3 and Model S) and 5% of Nissan Leaf vehicles.

### 2.5. OCPP Supervisor

From June 2021, the CEA has set up an IT architecture to control part of its EVCI. Twenty four (out of 80) charging points have been connected to an information system developed by the CEA, called SIGE for "Système d'Information pour la Gestion optimisée de l'Energie" in French (or EMIS for "Energy Management Information System" in English). This information system stores all the data necessary to control the charge of the EVs (data on EVs, data on the electrical network, characteristics of PV plants and charging stations, etc.). SIGE can also automatically download external data. For example, SIGE is connected to servers, in order to retrieve the PV production forecasts, and to PV plants, in order to retrieve the values of PV production. SIGE and the Diva stations communicate through a 3G connection by using the OCPP 1.6 protocol. The interactions of SIGE with the managers of the EVCI and the EV users are described after.

### 2.6. SIGE for the Managers of the EVCI and for the EV Users

SIGE offers a web user interface for the managers of the EVCI. It allows the operators not only to know the state of all the charging stations, but also to act on them. For example, the operator can stop the current charging session or restart it. The operator can also unlock a plug or set the maximum charging power. The web interface also provides statistics by user, by area, by charging station and by charging point.

The EV user must authenticate himself using the RFID badge issued by the CEA in order to be able to connect his vehicle to one of the charging points. This badge is first presented to the RFID reader of the Diva. The latter reads the ID number stored in the badge and the Diva station queries SIGE. If the badge ID is in the database and if the user participates in the experiments, SIGE sends a text message inviting him to fill in his forecast departure time (further noted $t_{dep}$) and the SOC of his car at the start of the charging process (further noted $SOC_{init}$). Without a response from the user, SIGE considers default values (i.e., $t_{dep}$ = 5:00 p.m. and $SOC_{init}$ = 25%). At the same time, SIGE sends the authorization of charge to the charging station while the user plugs his EV to the Diva terminal connector. The charging station then communicates with the vehicle via the cable, by carrier current, in accordance with the IEC 61851 standard, an international standard for electric vehicle conductive charging systems. Then, SIGE sends a charging test profile to the Diva constituting of two phases. The first phase, called the "discovering phase", involves charging the EV battery at its maximum power during a short period. This phase enables SIGE to estimate the maximum charging power of the EV, noted $P_{Max}$, but it also confirms to the user that his vehicle is electrically connected. Afterwards, in a second phase, SIGE controls the charge of the EV according to given objectives (see Section 2.7). When the charge is finished, SIGE sends a text message to the user, inviting him to disconnect his vehicle and to clear the parking place.

### 2.7. Control Algorithm

#### 2.7.1. Objectives

The CEA has developed a software module in SIGE, which controls the charge of the EVs that are connected to a set of charging stations. This controls processes according to given objectives and from all the data stored and retrieved by SIGE. The main objective is to satisfy the EV user, i.e., his EV battery is fully charged at his forecast departure time. The second objective is to maximize the self-production rate of the system constituted by 24 charging points and of 1.3% of the production of electricity produced by the Mégasol platform (160 kWp). The self-production rate is commonly defined as the value of the PV energy consumed when it is produced, divided by the value of the total consumption. However, in order to separate the effect of the EV charging control algorithm and the effect of the PV forecast algorithm, we decided not to consider the PV production in this definition but to consider the best forecast of this production, further called the "available power" and detailed in Section 2.7.4. Thus, we consider in the following that the self-production rate, SP, is defined as the value of the energy consumed when it is forecasted to be produced $E_{av}$, divided by the value of the total consumption $E_{cons}$. It is defined as the following equation:

$$SP = \frac{E_{av}}{E_{cons}} \tag{1}$$

#### 2.7.2. Charging Power Models

In the considered set-up, controlling the charge of an EV involves the control of its SOC by modifying over time the power that the station can deliver. The set of these maximum power values over time is called "SetPoints". The function that links the SetPoints and the SOC is very complex. It depends on the maximum power accepted by the car when it is charging with an AC connection. This parameter depends on the car model, on the size of its onboard charger, on the type of connection with the charging station and on the cable that is used to connect the car to the station. It also depends on the external

temperature and on the traction battery temperature. Besides, the user may use a software (generally an application on a smartphone) that may also limit the value of $P_{Max}$. In the experiments, the participants are encouraged to deactivate such external control. In the following, we consider that $P_{Max}$ is constant and that it is equal to the value measured during the "discovering phase". The energy that can be stored in the traction battery, further called "Capacity" and noted $E_{Max}$, is also a key parameter that depends on the vehicle model and on options. For example, a Tesla Model 3 may have four sizes of battery: 55, 62, 75 and 82 kWh. In our case, we consider that Capacity is constant and equal to the value declared by the participants. For the cars of people not participating in the experiments, Capacity is considered as the maximum size of the battery for the EV model. Other parameters exist, such as the number of phases that are used to charge the car. We consider public information to obtain these data. In general, if the onboard charger is less than 11 kW, it is one phase, otherwise, it is a three-phase charger. However, there are exceptions: for example, the Seat Mii charge at 7.4 kW with a two-phase charger.

### 2.7.3. Control Principle

The planning algorithm is executed at each start of a new charging session but also periodically every 10 min. Let us consider that $t_0$ is the current time, that the 24 h after the current time are discretized by time step of $\Delta t = 10$ min and that those points in time are noted $t_n$. As, in our set-up, it is not possible to directly measure the SOC of the cars, the planning algorithm first estimates the SOC of all the EVs at current time $t_0$. These estimations are based on the initial values of the SOC and on the measurements of the energy meters of the charging stations (see details in Section 2.7.5). Then, for all $t_n$, the control algorithm follows two steps:

1.  The available power values are calculated, as detailed in Section 2.7.4.
2.  These values are allocated amongst the cars that are connected and that are waiting to be charged (i.e., their SOC estimate is lower than 100%). The allocating process is based on a basic scheduling mechanism, i.e., "earliest deadline first" [20], which was adapted to achieve our objectives. In our context, we consider that the "deadline" of a car, also called "lead-time", is the difference between the time before departure ($t_{dep}$) and the time needed to recharge the battery without a charging control ($t_{end}$). The computation of this lead-time is detailed in Section 2.7.6. Then, the planning algorithm estimates the lead-time of all the EVs. The available power is first allocated to the EVs that are not under control (i.e., EVs in set "Uncontrolled") and to the cars that have a negative lead-time. The maximum charging power is allocated to each of these cars. Then, the remaining power is distributed in the inverse order of the lead-time. The maximum charging power is allocated to the cars that have the smallest lead-time. If there is not enough available power for the cars that have the highest lead-time, they do not recharge during this time step but they will be charged during the next ones. The powers allocated to each EV, for each time step, are the SetPoints that are sent over the 3G network to the charging points.

The following simple example enlightens the principle of the control algorithm. Let us consider an EV owner, not participating in the experimentations, who connects his car (a Zoé with a battery capacity of 50 kWh) to the EVCI. Since SIGE has no information about the initial SOC of the car battery, it takes $SOC_{init} = 25\%$ as a default value. Thus, SIGE considers that around 40 kWh are needed to fully charge the battery. In this example, we assume that there is enough available power to supply the car battery at $P_{Max}$ (here 22 kW) during the time needed to fully charge the battery, i.e., about 1 h 50. In such a case, the planning algorithm send the associated set points (i.e., 22 kW during 1 h 50) to the station. Then, SIGE estimates the charging power as explained in Section 2.7.5. Two cases arise during the charge:

1.  The car may either stop charging because it is full prematurely (i.e., before 1 h 50 of charge). SIGE has underestimated the real SOC. It may happen when the real initial

SOC is greater than 25%. In that case, the set points are null power values and the charging power is allocated to another car.

2. Or it may continue to charge even after 1 h 50 of charge. SIGE has overestimated the real SOC. It may happen when the real initial SOC is less than 25%. In that case, the set points do not change until the charge stops.

We have simulated the effect of this control algorithm on the historical charging session data presented in Section 2.3. This simulation step allowed confirmation of the choice of the different parameters and proved the effectiveness of the control algorithm.

### 2.7.4. Available Power

Figure 2 illustrates how the power considered as available in the future for the charge of the EVs is computed. It shows the solar production data of 1 November 2021. First, the PV production forecasts, based on weather forecasts, are retrieved every two hours from the web server of the National Oceanic and Atmospheric Administration in the USA. These data are called "rawPowerForecast" and are reported as orange dots in Figure 2. Second, the power production of the PV plant considering that the sky is clear, is computed through a homemade software based on the public software library PV_LIB [38]. These data are called "clearskyPower" and are reported as a dark blue line. This curve is bell-shaped because solar production increases during the morning until it reaches solar noon (time when solar production is at its peak), then decreases at the same rate as it increased a few hours earlier. Third, a "bell-shaped" interpolation of the "rawPowerForecast" is computed. These values are called "powerForecast" and are reported as a black line. The power measured on the PV plant is collected with a System Control And Data Acquisition (SCADA) and is transmitted over the 3G network. These data are called "Power" and are reported as a blue line. From all these data, SIGE computes the "correctedPowerForecast" represented by a green line. Roughly speaking, these last data are equal to the "powerForecast", unless the previous values (i.e., in the near past) of "powerForecast" are too far from the previous values of "Power". In that case, "correctedPowerForecast" is computed through "Power" and through the persistence method (i.e., the weather in the near future is considered the same as the weather in the near past). Finally, the power considered in the near future as available for the charge of the EVs is the "correctedPowerForecast".

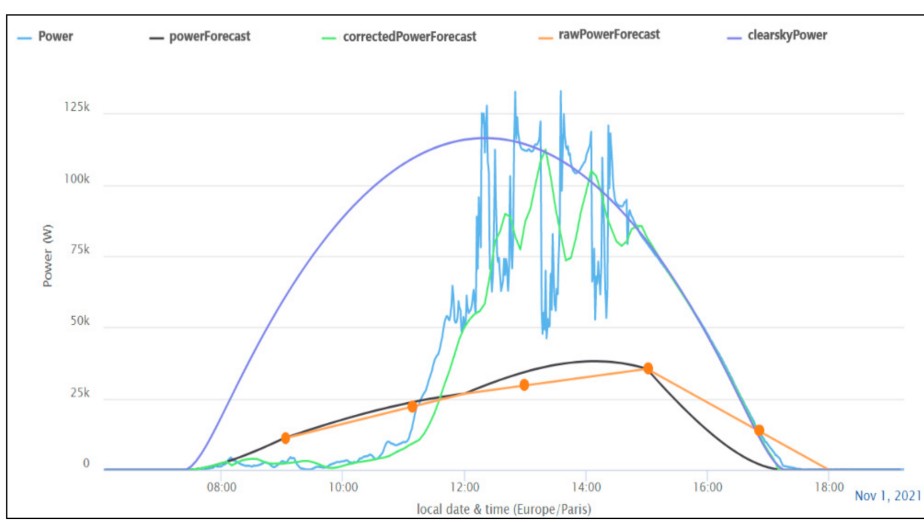

**Figure 2.** Illustration of the computation of the available power.

### 2.7.5. SOC Estimation

The value of the SOC estimated at a given time $t_0$ is further noted $SOC_{estimate}(t_0)$. This estimation is computed using the initial value of the state of charge $SOC_{init}$ and using the values of the energy meter, read every dt equal to 2 min. For many charging stations, it is the minimum time to retrieve data without any problem of synchronisation or timestamp

issue. The power withdrawn by the car $P_w(t_0)$ is first estimated from the values measured by the energy meters at the considered time $t_0$, noted $E_w(t_0)$, and Q dt minutes before $t_0$, noted $E_w(t_{-Q})$. The power $P_w(t_0)$ is computed according to the following formula:

$$P_w(t_0) = \frac{E_w(t_0) - E_w(t_{-Q})}{Q \times dt} \tag{2}$$

Q is a parameter chosen equal to 3 to obtain an accurate value of the power as quickly as possible (i.e., every 6 min). We consider that if the car does not withdraw energy during the six minutes before $t_0$, the measured withdrawn power $P_w(t_0)$ is equal to zero. This phenomenon occurs because the car battery is fully charged, and then the $SOC_{estimate}$ is equal to 100%.

If $P_w(t_0)$ is not equal to zero, we compute $SOC_{temporary}(t_0)$ according to Equation (3). In this equation, we consider $t_{init}$ being the time at the start of the charging process, and Y being the yield of the charging process (supposed to be equal to 95%).

$$SOC_{temporary}(t_0) = SOC_{init} + Y \times 100 \times \frac{E_w(t_0) - E_w(t_{init})}{E_{Max}} \tag{3}$$

If the car keeps charging (i.e., $P_w(t_0) > 0$) while $SOC_{temporary}(t_0)$ equals a threshold value (typically 99%), we consider that the value of $SOC_{temporary}$ is overestimated compared to the real SOC. In that case, without any other information about the real SOC, we consider that $SOC_{estimate}$ stays equal to the threshold value. In other cases, we consider that $SOC_{estimate}$ is equal to $SOC_{temporary}$.

### 2.7.6. Lead-Time

As explained above, the lead-time is the difference between the time before departure and the time needed to fully charge the battery without a charging control. This lead-time has to be estimated at each time step $t_n$. The set points from $t_0$ and $t_{n-1}$ have been computed by the previous iteration of the planning algorithm. We note $SetPoint(t_p)$ the power set point computed for the period $[t_p; t_p + \Delta t]$. During this period, we consider that the car charges at the constant value $SetPoint(t_p)$. Thus, at $t_0$, the value of the $SOC_{future}$, in a future point in time $t_n$, is calculated according to the following formula:

$$SOC_{future}(t_n) = \max\left( 100,\ SOC_{estimate}(t_0) + \frac{100 \times Y}{E_{Max}} \times \sum_{p=0}^{p=n-1} SetPoint(t_P) \times \Delta t \right) \tag{4}$$

Then, from time $t_n$, we consider that the EV charges at $P_{Max}$ until its battery is fully charged. Given this $SOC_{future}$ value, we are able to compute the point in time $t_{end}$ such the battery is full when charged with the following formula:

$$t_{end}(t_n) = t_{init} + (100 - SOC_{future}(t_n)) \times \frac{E_{Max}}{100 \times P_{Max}} \tag{5}$$

The lead-time is the difference between the time before departure $t_{dep}$ and the time $t_{end}$ to charge the battery at $P_{Max}$.

$$LeadTime(t_n) = t_{dep}(t_n) - t_{end} \tag{6}$$

### 3. Results

*3.1. Preliminary 1: Data Selection*

The following experimental data were recovered between 1 October 2021 and 1 February 2022 inclusive (124 days in total). We consider that 73 days give exploitable results. The other days are either without any EV charging session (34 days) or with measurement errors or communication or hardware breakdowns (17 days). During these field tests, 887 charging sessions were recorded in total, and, altogether, 12.4 MWh were transferred

to the EVs. The battery of the EVs were all charged before the departure time forecasted by the users, except in very few cases (mainly short sessions of 1 h or 2 h). In these cases, the users were informed by email or phone that there were issues with their charging sessions.

In order to estimate the efficiency of the control algorithm, we compared two self-production rates. The first one, SP, is computed from the available power and from the energy consumption measured by the charging stations, as explained in Section 2.7.1. The second rate is computed in the same way, except that we consider the energy consumption as the simulated consumption of all the EVs, under the assumption that none of them are controlled. In other words, we quantify the total energy in the case where the EVs charge at their $P_{Max}$ from the start of the charging sessions until their batteries are fully charged. The associated self-production ratio is denoted "Uncontrolled Self-production ratio" or $SP_u$.

Figure 3 displays a scatter plot with the self-production ratio SP on the Y-axis and the uncontrolled self-production ratio $SP_u$ on the X-axis, computed both on the same days. As the points are clearly above the line Y = X, this scatter plot clearly shows that the control algorithm globally increases the self-production ratio. A small (resp. great) difference between SP and $SP_u$ corresponds to a minor (resp. large) increase in the self-production ratio thanks to the control algorithm.

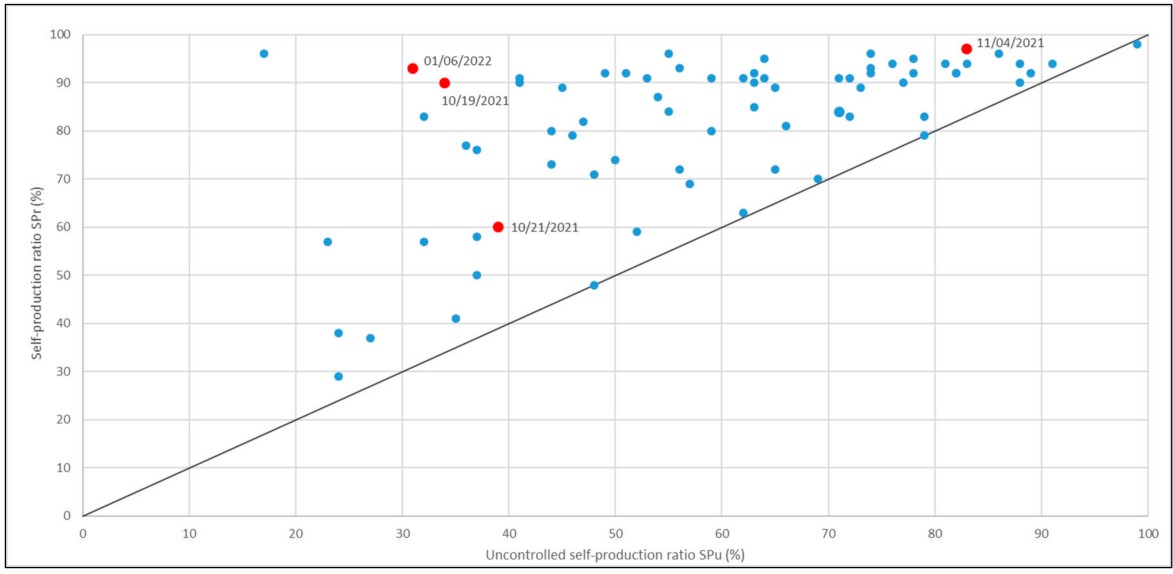

**Figure 3.** Distribution of the self-production ratio in both controlling scenarios on a scatter plot (red dots correspond to days that will be described in the next paragraphs).

The days associated with the red dots in Figure 3 are chosen according to their different locations on the plot (on the middle, in the upper right corner, and at top left). Indeed, these red dots represent cases that will be outlined in the following paragraphs:

- On the middle: low values for both SP and $SP_u$
- In the upper right corner: high values for both SP and $SP_u$
- At top left: high value for SP and low value for $SP_u$

### 3.2. Preliminary 2: Description of Figures 4–7

Figures 4–7 are partly made up of three curves related to PV production, as described in Section 2.7.4:

- The blue curve corresponds to the "ClearskyPower".
- The red curve represents the "CorrectedPowerForecast" which is considered as the available power.

- The green curve represents the measured PV production noted as "Power". The measurement system of these data breaks down sometimes, as can be seen in Figures 4–7 (when the green line is flat or missing).

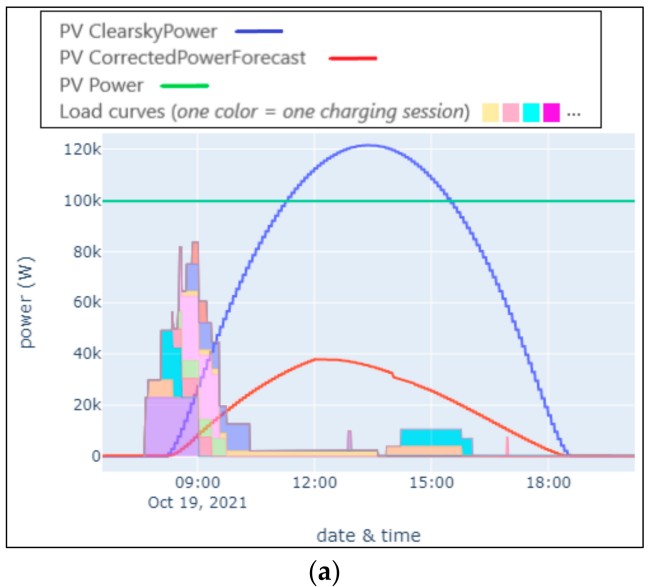

(**a**)

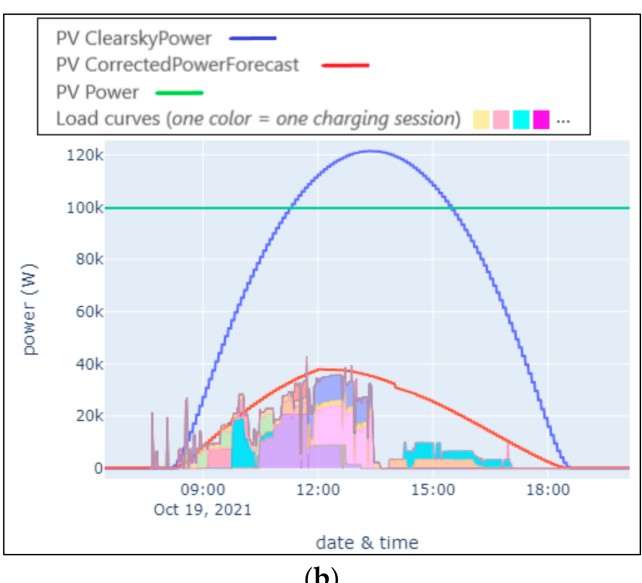

(**b**)

**Figure 4.** Results of 19th October 2021: (**a**) Stacked energies without control. The self-production rate $SP_u$ is equal to 34%; (**b**) stacked energies with control. The self-production rate SP is equal to 90%.

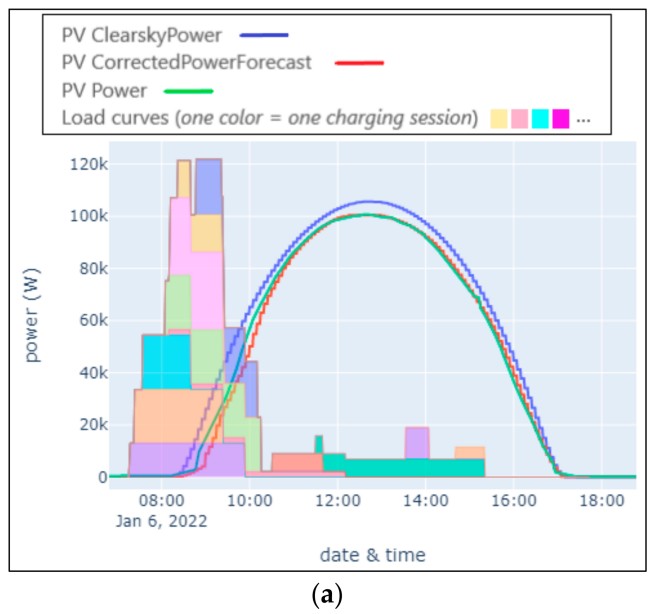

(**a**)

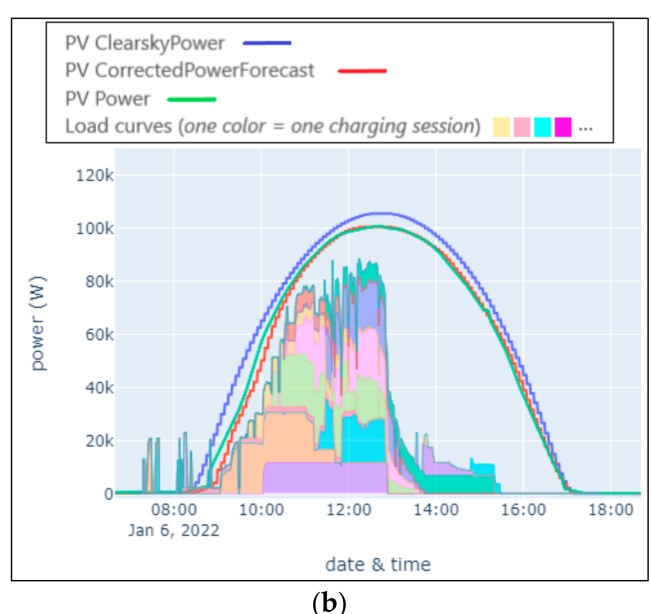

(**b**)

**Figure 5.** Results of 6th January 2022: (**a**) Stacked energies without control. The self-production rate $SP_u$ is equal to 31%; (**b**) stacked energies with control. The self-production rate SP is equal to 93%.

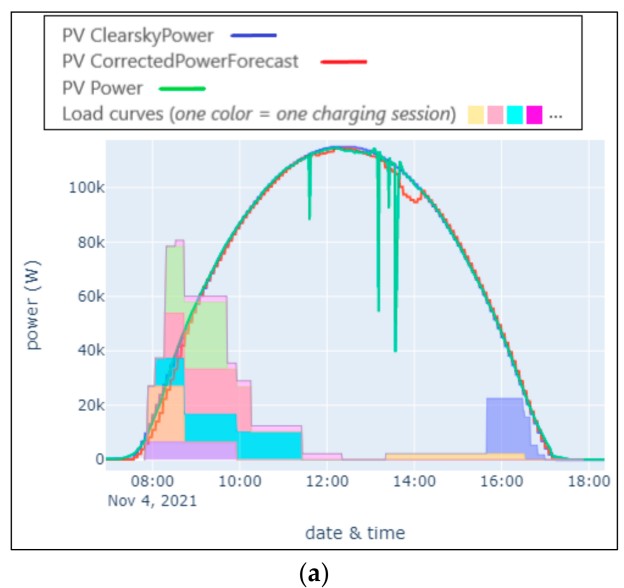
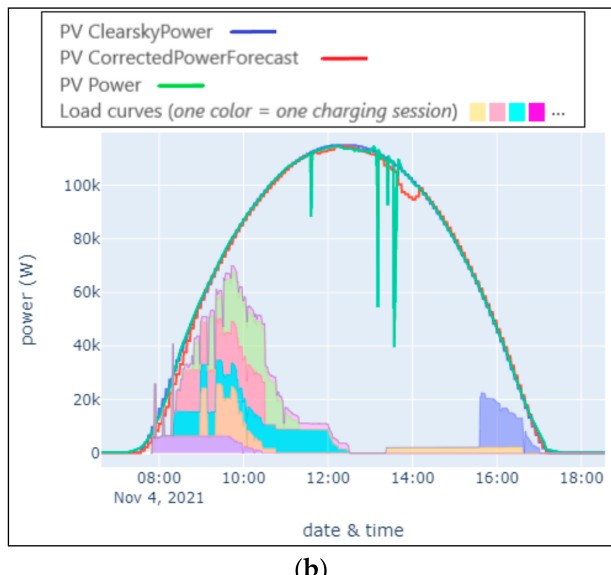

(**a**)     (**b**)

**Figure 6.** Results of 4th November 2021: (**a**) Stacked energies without control. The self-production rate $SP_u$ is 83%; (**b**) stacked energies with control. The self-production rate SP is 97%.

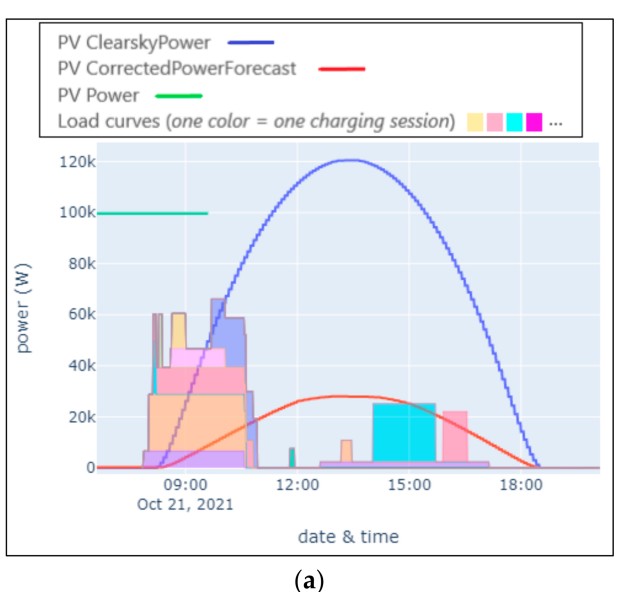
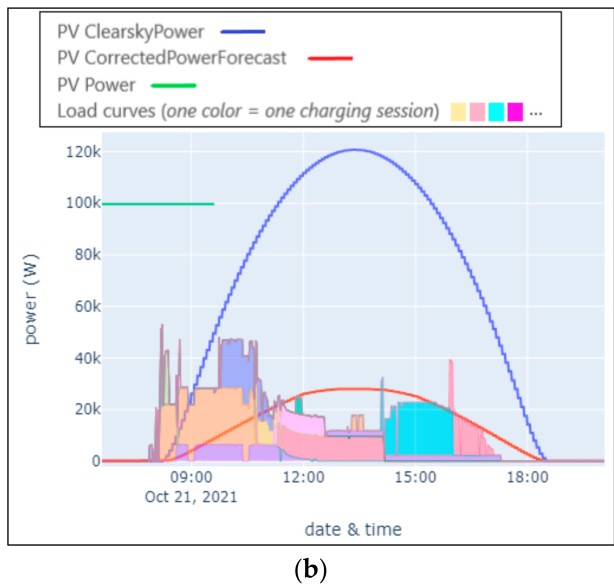

(**a**)     (**b**)

**Figure 7.** Results of 21st October 2021: (**a**) Stacked energies without control. The self-production rate $SP_u$ is equal to 39%; (**b**) stacked energies with control. The self-production rate SP is equal to 60%.

Those figures also depict the charging sessions over time for an entire day. The graphical representation of the power withdrawn by the car over time (estimated as explained in Section 2.7.5) is further called the "load curve" of the charging session. The area under this load curve corresponds to the energy transferred to the car. In the following, we associate each such area (and so, each charging session) with a different color. At last, we stack all these areas on top of one another. Thus, the top of the stacking at a given time corresponds to the total charging power at this time.

Each figure is composed of two graphs, (a) and (b). They respectively correspond to the load curves that would have been obtained without control, and the measured load curves obtained with the control algorithm. In each case, we also report the total energy transferred to the EVs and the PV production during the day.

### 3.3. The Control of Charge Increases Drastically the Self-Production Ratio

Figure 4 represents the results obtained on 19 October 2021. During this cloudy day, 17 charges occurred, for a total energy consumption of 141 kWh and for a total PV production of 221 kWh.

Figure 4a shows that the charges, without any control, would have been carried out mainly between 7:30 and 10:30 a.m. Figure 4a shows that the cumulative load curve is located rather above the red curve, which means that the self-production rate is low (in that case, it is equal to 34%). On the contrary, Figure 4b shows that the loads are happening mainly below the red curve, which suggests that the self-production rate is high (here 90%).

Figure 5 represents the results obtained on 6 January 2022. On this particular sunny day, 15 charges occurred, for a total energy consumption of 227 kWh and a total of forecast PV production of 551 kWh. This particular day, the self-production rate increase from 31% to 93% thanks to the control algorithm.

These results emphasize that the charging control may increase drastically the rate of self-production during both sunny days and cloudy days.

### 3.4. The Self-Production Rate Can Be High, Even without Control

Figure 6 illustrates the results obtained on 4 November 2021. During this sunny day, nine charges occurred, for a total energy consumption of 165 kWh and for a total PV production of 715 kWh. This particular day, both self-production rates, with or without control, were high (here 97% and 83% respectively). This is because the charging sessions start mostly at the beginning of the working day, as explained in Section 2.3. Thus, most of the charging sessions take place in the morning, when the PV production is increasing. This example emphasizes that the self-production rate may be high, with or without control.

### 3.5. The Self-Production Ratio May Be Relatively Low, Even with Control

Figure 7 illustrates the results obtained on 21 October 2021. During this sunny day, 19 charges occurred, for a total energy consumption of 214 kWh and for a total PV production of 174 kWh. This particular day, the self-production rates, with or without control, are both relatively low (here 60% and 39% respectively). This is because two EV users (one Tesla Model S user, and one Zoé user), who did not give us control over the charge of their EVs, charged their EVs at high power, i.e., 22 kW for both cars, early in the morning. Such a result stresses that the users' behavior strongly influence the results.

## 4. Conclusions and Perspectives

The experiments conducted on CEA Cadarache site benefits from the city-size of the research center and from its CPO, eMSP and DSO roles. They involve nearly 100 out of the 376 EV users and more than 40 different car models. The objectives of these experiments were to fully recharge the battery of the users before their forecast departure time and to maximize the self-production rate of the system, which is constituted by 24 charging points and of part of the production of a PV plant.

The first goal was reached, except in very few cases. We consider that the second goal was partially reached. Indeed, we compared the self-production rates obtained with or without control. Our analysis showed that the charging control always increases this ratio. Some days, especially during grey ones, the ratio drastically increased (we gain sometimes more than 60 points of percentage). However, there are several days, especially sunny ones, when the increase was rather low. We also showed that, sometimes, the ratio was rather low, even with control. From our point of view, this is mostly due to the users that do not give us control over the charge of their EVs. One way to improve the ratio is to convince these users to change their mind. This could be done, for example, by organizing "solar charging" contests, by communicating intensively in the Cadarache center on obtained experimental results or by offering a financial incentive. Another way would be to enforce smart charging for all EVs but the risk is to reduce the users' satisfaction and their involvement in the experiments. We believe that such an option could be counterproductive because this

involvement is mandatory to get access to key data such as the initial SOC and the forecast time of departure.

As perspectives, we plan to improve our planning algorithm in three main ways. First, we plan to improve the models that relate the value of the set points and the power withdrawn by the car, as suggested in [39]. These models are used during the allocation of the available power. In the current implementation of our algorithm, any error in the model translates into in a loss of power. In other words, this loss of power is not allocated to other cars. To improve such models, we expect to increase the number of charging points, of users, and thus, of car's models. We also forecast to monitor other physical variables useful for the modeling, such as the SOC and the battery temperature. Second, we consider turning the planning algorithm into an optimization problem and then to apply, for example, Mixed-Integer Linear Programming (MILP) methods. At last, the retrieved data could serve as a basis for defining other Key Performance Indicators, such as the state of health of car battery, the occupancy rate of each charging station or the amount of power not extracted from the grid thanks to the charging control. This latter could be used to quantify the economic benefit of EV solar charging. The satisfaction of users is also a key issue that could be estimated. This is especially true when users are fully involved and are accustomed to giving their charging features at the beginning of the charging sessions. In a more long-term horizon, we plan to assess the EV capability to provide flexibility services to the electricity network and to estimate the benefit for the grid.

**Author Contributions:** Conceptualization, B.R. and S.G.; methodology, B.R.; software, S.G.; validation, B.R., L.M., G.V., S.G. and A.M.; formal analysis, B.R. and L.M.; investigation, B.R. and L.M.; resources, S.G., B.R. and L.M.; data curation, S.G. and B.R.; writing—original draft preparation, B.R. and L.M.; writing—review and editing, G.V., A.M., S.G., L.M. and B.R.; visualization, S.G.; supervision, B.R.; project administration, B.R. and G.V.; funding acquisition, G.V., A.M. and B.R. All authors have read and agreed to the published version of the manuscript.

**Funding:** This research was partially funded by RTE and CEA. This work was supported by ADEME France, project PV2E_Mobility, grant number #1905C0043, by the French National Program "Programme d'Investissements d'Avenir-INES.2S" under Grant ANR-10-IEED-0014-01.

**Institutional Review Board Statement:** Not applicable.

**Informed Consent Statement:** Not applicable.

**Data Availability Statement:** Not applicable.

**Acknowledgments:** We thank the CEA Cadarache director, J. Vayron, for his interest in the experiments. We would like to show our gratitude to the logistics department of CEA Cadarache, especially V. Giroud, P. Caron, G. Demeulemeester and L. Angelini for making charging stations from the EVCI available for our experiments. We would also like to thank the technical services for their maintenance operations and our colleagues F. Mezzasalma and A. Blaise for their technical assistance. We thank our colleagues from INES/CEA Grenoble who provided expertise for the software development. We are immensely grateful to all EV owners of CEA Cadarache, who are volunteer experimenters, and without whom the experiments would not exist. Thanks to them for enriching the experimental feedback. Finally, we would like to express our gratitude to Rowena Mathew as English-speaker reviewer and to the anonymous reviewers for their fruitful comments.

**Conflicts of Interest:** The authors declare no conflict of interest.

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
