# Peer review of "Solar Charging of Electric Vehicles: Experimental Results"

_applsci, doi:10.3390/app12094523_

Round 1

Reviewer 1 Report

The Authors should carefully review the paper and check for typos and grammar flaws. Also, use words such as "recharge", "charge", "fully charge", instead of "fill up" or "fulfill". 

In general, all Figures should be reviewed. In particular:

  • Figure 1 is unreadable and labels are in French...
  • Figure 2 is unreadable. It seems there is a subplot at the bottom, what is it?
  • Figure 3 should not have a title which is a repetition of the figure caption.
  • Figures 4, 5, 6, and 7 need a legend to explain what the bell-shaped red, green, and blue curves represent. Can you better explain how the load curves are represented? If charging stations are max 22 kW, why we see higher charging powers?

At lines 133 and 134, "power consumption" should be replaced with "energy consumption". Power and energy are also confused in other cases throughout the manuscript, please correct.

At line 217, "This self-production rate is commonly defined as the value of the PV 217 energy consumed when it is produced, divided by the value of the total consumption." Can the authors use a mathematical equation to define the self-production rate?

At line 295, "This data is called “clearskyPower” and is reported as a yellow line. Third, an interpolation of the “rawPowerForecast” with a curve that has the same shape than the “clearskyPower“ is computed." What the Authors mean by "same shape"? In the same paragraph, it is not clear why there is a need to forecast the solar power if the actual power can be measured directly. Can the Authors clarify?

At line 314, the choice of values "six minutes" and "Q a parameter chosen equal to 3" need to be further explained.

Reviewer 2 Report

In the paper, the authors presented a supervision system that controlled the electric vehicle charging of employees of CEA Cadarache research centre. The set-up and the experiments were demonstrated in detail. However, there are some major problems to be solved before publication.

  1. Some important experimental results or conclusions should be included in Abstract.
  2. The text in Figure 1 is unreadable, please make it clear enough for better readability. please transform the axis title in English.
  3. Page 4, line 147: “As at 1st February, 376 RFID badges were issued.”, please add the year after the date.
  4. Page 9, Line 375-377: the authors declared that the days associated with the red dots in Figure 3 are chosen according to their different location on the plot (at the bottom left, in the upper left corner, and at top right). The description makes readers confused. Why to choose these days? Are there any meaningful reasons to choose these days? If so, the authors should give detailed information.
  5. In Figure 4, what do the red curve as well as the blue curve, and different color bars represent? These information should be shown in Figure 4, not just in text. The same change should be made in Figs 5-7.
  6. In Figure 4, the blue curve exceeds the Y-axis scale. Please explain the reason.
  7. Page 11, line 415: there may be some mistakes. “Thus, most of the charging sessions take place in the morning, when the PV production is significant”. However, as is well known, the PV production is significant at noon.
  8. The authors conducted some experiments and achieved certain amounts of results. So, what kind of guidance this paper could provide for the arrangement of future EV solar charging?

Reviewer 3 Report

The paper provides interesting experimental results.

It should be improved by introducing some theoretical support to experimental data and with some more detail on practical derivation coming from the experimental results in addition to those already provided.

As an example the impact on the grid of the proposed PV charging stations, and or the benefit derived from the grid support to the PC charging stations.

Which are the action that could be considered to achieve the goals in those cases the, as paper states, they are not achieved?

Any model to support the proposed actions showing their effectiveness?
